# Fooling 3D Face Recognition with One Single 2D Image

## ABSTRACT

3D face recognition is subject to frequent spoofing attacks, in which 3D face presentation attack is one of the most notorious attacks. The attacker takes advantages of 3D scanning and printing techniques to generate masks of targets, which has found success in numerous real-life examples. The salient feature in such attacks is to obtain 3D face models through 3D scanning, though relatively more expensive and inconvenient when comparing with 2D photos. In this work, we propose a new method, DREAM, to recover 3D face models from single 2D image. Specifically, we adopt a black-box approach, which recovers 'sufficient' depths to defeat target recognition models (e.g., face identification and face authentication models) by accessing its output and the corresponding RGB photo. The key observation is that it is not necessary to restore the true value of depths, but only need to recover the essential features relevant to the target model. We used four public 3D face datasets to verify the effectiveness of DREAM. The experimental results show that DREAM can achieve a success rate of 94% on face authentication model, even in cross-dataset testing, and a success rate of 36% on face identification model.

## CCS CONCEPTS

• Security and privacy → Systems security.

## KEYWORDS

3D face recognition, Twin deep networks, RGB-D images, Black-box attack

## 1 INTRODUCTION

Face is one of the most important human biometrics, which has been widely used in authentication (verification) or identification systems. In such systems, there are two stages, i.e., registration and recognition. During registration, a user registers his or her face to the systems (as template). During recognition, a face will be compared with all templates stored in the system, to determine whether it matches with one of registered face or not. The face (and the corresponding individual) is identified and recognized as legitimate if there is a match. Although 2D face recognition has been widely used, due to lack of depth or distance information, the face expression is incomplete, often rendering inefficiency under dim lights and variants of poses and positions. In recent years, face recognition has gradually shifted from 2D to 3D, as the prices of depth cameras have been considerably decreased.

*ACM MM, 2024, Melbourne, Australia*
© 2024 Copyright held by the owner/author(s). Publication rights licensed to ACM.
ACM ISBN 978-x-xxxx-xxxx-x/YY/MM
https://doi.org/10.1145/nnnnnnn.nnnnnnn

There have been a variety of spoofing or adversarial attacks against 3D face recognition systems. Among them, 3D presentation attack has been extensively studied, which refers to use 3D scanning and printing technologies to obtain a real size 3D reproduction of the victim's face [7, 37, 40] so as to fool 3D face recognition systems. Unfortunately, building 3D face models requires specialized 3D scanning equipment, which incurs high cost (e.g., a portable optical 3D scanner GOM TRITOP costs RMB 50,000 [34]). In contrast, it is much cost effective to obtain 2D images instead, for instance, from video conference and surveillance camera etc. Hence, in this work, we seek to answer the question whether it is possible to build a 3D model from the captured 2D image to fool 3D face recognition systems.

A direct way of launching 3D face spoofing attack with 2D facial images is exploiting 3D-from-2D reconstruction technologies to accurately recovery the corresponding 3D facial presentation. These technologies can be divided into three strategies, i.e., *photometric stereo* [8, 26, 31], *statistical model fitting* [4, 17, 36] and *deep learning* [32, 46, 58]. Photometric stereo estimates the local orientations of the face surface from a sequence of facial images (typically three or more) of the target person, obtained from the same viewpoint and under varying illumination, which are then integrated to establish the face geometry. Statistical model fitting requires massive 3D faces of different persons in order to obtain a statistical 3D facial model to which small modification is applied according a 2D facial image to be reconstructed. Given the fact that fine-tuning is achieved through a limited set of model parameters, it may not be possible to perfectly restore local detailed geometric features, resulting in poor attack performance. Learning the 2D-to-3D mapping through deep neural network (DNN) is a promising way. Some design a DNN model to predict 3D facial models according to the input 2D face images, the training of which requires huge amount of 3D facial scans and is not always feasible [33]. Others learn the depth of 3D scenes from pictures[39, 55]; however these methods only recover rough depths rather than precise local geometries required for facial recognition. As a result, for an attacker who holds a single image of its victim and a small set of 3D faces, there is no successful way to launch spoofing attack against 3D face recognition systems.

In this paper, we propose a novel 3D face spoofing attack, Depth Recovery Attack Method (DREAM). In our attack, an attacker holding only one 2D face image establishes a 3D face which can pass the face recognition of the target system. For example, one possible attack scenario is that an attacker tries to unlock the victim's mobile phone (within the number of consecutive unlock attempts allowed by the system) while he or she leaves temporarily. DREAM is inspired by the observation that a face recognition system commonly refers to a deep neural network extracting and comparing key features between the registered template faces and the probe face. This implies that 1) *DREAM does not necessarily need to restore the entire 3D face precisely, but only needs to reconstruct a model sufficient to defeat the target face recognition system, which means those key features relevant to face recognition are more important and*

should be given more attention. 2) 1) *It is possible to learn key facial features 'remembered' in the recognition model by interrogating it.*

We emphasize that achieving our attack is however challenging due to the following two reasons. Firstly, the attacker can only access the target commercial device in a black-box manner, and the number of consecutive interrogations is limited, e.g., *5*. This means that the knowledge about 3D facial templates could be obtained from the target device is very limited. Secondly, black-box accessing implies that the key features extracted for facial feature comparison cannot be obtained, making it difficult to determine the priority of different local features in geometric reconstruction. To overcome the above challenges, DREAM is composed of two stages, i.e., *zero-knowledge pre-training of depth generator* and *depth fine-tuning with target model.* The attacker uses a rough 3D face learned in the first stage to query the target device so as to refine it, which guarantees a high attack success rate within the permitted number of consecutive interrogations.

Specifically, in the first stage, a naive way to supplement the facial depth according to a face image is to train a generative adversarial network (GAN) [13] with a 3D face dataset held by the attacker. Notice that the dataset can be publicly accessible and is irrelevant to those registered users of the target device. However, such a solution leads to a poor attack success rate, because GAN is not able to be aware of key depth information solely. Thus, we use a pre-trained agency face recognition model, and registers those 3D faces to it. The attacker can obtain an agency model in many ways, such as downloading the same (or similar) recognition applications or using a device of the same type as the target device. Then the agency model is responsible of supervising the facial depth reconstruction. Meanwhile, we introduce an attention module in the generator to distinguish key features, and design dual-contrast loss [52] which enables the generator to gain stronger capability from small-shot learning. In the second stage, the attacker uses the 3D face obtained in the first stage to query the target device, leading to two results. The first is passing the facial recognition directly, i.e., attack success. Otherwise, the attacker uses the device response, e.g., similarity score, to optimize the input of the GAN network (i.e., random noise), resulting in an updated 3D face will be used in the next round of query. The above process is repeated until the attack success or the maximum number of queries is reached.

We launch our DREAM against both 3D face authentication (1v1) and identification (1vn). We choose the widely used Siamese architecture [6] as the target authentication model, and Led3D [35] and the architecture proposed by Uppal *et al.* [47] as the target identification models, respectively. Three public 3D face datasets, Pandora [5], RGB-D Facial Dataset under Pose Variation [20] and Texas Database [16], are used for authentication, and Lock3DFace [54] is for identification. Extensive experimental results show that the attack success rates of DREAM are 94.73%, 85.71% and 88.50% within 5 attempts against face authentication model, and 36.36%, 89.32% within 5 attempts against face identification models.

## 2  RELATED WORK

In this section, we briefly review 3D face recognition system, 3D face reconstruction, and GAN.

### 2.1  3D Face Recognition System

Face recognition, as a biometric technology, has gained widespread applications due to its ubiquity and non-invasiveness. It is extensively utilized in various fields such as security, commerce, healthcare, and robotics applications [25, 41, 56]. Existing face recognition techniques can be broadly divided into 2D and 3D face recognition techniques according to the data modality [15]. Thanks to the low price and wide availability of 2D image acquisition devices, most research efforts and commercial developments have focused on 2D face recognition. However, with the advancements in 3D sensing devices (such as Microsoft Kinect, Intel RealSense) and computing devices (such as GPU), 3D face recognition is gradually entering everyday life, for example, Apple Face ID [1]. 3D face can be represented by data formats such as RGB-D image, point cloud and mesh. Some work [20, 47, 48] propose taking RGB-D image pair as input and extracting RGB and depth features for face recognition. Some cloud platforms [3, 44] also use RGB-D image pair as input, but use the depth for liveness detection and extract RGB feature for face comparison. Led3D [35] only take face depth as input for face recognition. PointFace [19] directly extracts features using face point cloud and calculates the similarity with template point clouds for discrimination.

3D face recognition has many advantages over 2D face recognition [21, 22, 28]. For example, (1) 3D data contain sufficient facial geometric information without requiring any projection from 3D physical space to 2D imaging plane, so it an provide more discriminating features for face recognition; (2) 3D data is insensitive to changes in pose, illumination, and expression. Therefore, face recognition systems that use 3D data will be more robust to changes in the surrounding environment and better suited to real-world situations. (3) 3D face recognition systems have higher security, it use depth information for liveness detection and naturally resistant to common 2D printing and replay attacks [3, 42, 44].

Face spoofing attack can be divided into 2D and 3D spoofing based on the attack method [18]. 2D spoofing typically involves using printed photo or electronic screen, while 3D spoofing often involves the use of face masks. Naive 2D spoofing attacks cannot pose an effective threat to 3D face recognition systems, but when combined with optical attacks, they can compromise 3D face recognition systems. DepthFake [51] estimates the 3D depth information of a target victim's face from his 2D photo. Then, DepthFake projects the carefully-crafted scatter patterns embedded with the face depth information, in order to empower the 2D photo with 3D authentication properties. Experiment show that DepthFake can spoof multiple commercial SDKs and devices. 3D spoofing attack and morphing attack usually use meticulously crafted facial mask as tool, with the mask either closely resembling the victim or being meticulously calculated through adversarial attack. Singh *et al.* [40] successfully deceived the 3D face recognition based on PointNet in digital attack scenarios by morphing the 3D information of two individuals. Li *et al.* [30] design end-to-end attack algorithms to generate adversarial illumination for 3D faces through the inherent or an additional projector to produce adversarial points at arbitrary positions. They successfully attacked point-cloud-based and depth-image-based 3D face recognition algorithms in both digital and physical worlds.

## 2.2 3D Face Reconstruction

Recently, incorporation of 3D data into face analysis and its applications attract a lot of attention. Despite providing a more accurate representation of the face and 3D sensor devices become more available, 3D facial images are more complex to acquire than 2D pictures and publicly available datasets are not on the same order of magnitude in terms of the number of images and subjects. As a consequence, great effort has been invested in developing systems that reconstruct 3D faces from an uncalibrated 2D image. However, as mentioned above, the 3D-from-2D face reconstruction problem is ill-posed, thus priori knowledge is needed to restrict the solutions space. According to [33], 3D face reconstruction can be divided into three types based on the method of adding prior knowledge, namely, statistical model fitting, photometry, and deep learning. The deep learning strategy became the state-of-art in the past few years because of better deep learning architectures and algorithms, it can reconstruct finer details, even animatable details. Lei *et al.* [27] present a novel hierarchical representation network (HRN) to achieve accurate and detailed face reconstruction from a single image, which implement the geometry disentanglement and introduce the hierarchical representation to fulfill detailed face modeling. Feng *et al.* [11] present the first approach that regresses 3D face shape and animatable details, because the proposed method introduces a novel detail-consistency loss that disentangles person-specific details from expression-dependent wrinkles. Due to the ability to restore high-fine details, 3D face reconstruction can be used to create a 3D mask of the victim's face for 3D presentation attack [9]. Marcel *et al.* [38] propose a novel template inversion attack against 2D face recognition systems, which is the first work on 3D face reconstruction from facial templates to reconstruct facial texture details.

## 2.3 GAN

GAN is one of generative models that is capable of learning generative tasks in semi-supervised or unsupervised application scenarios. It consists of two parts, i.e. generator and discriminator, the generator is mainly used to learn the distribution of real images so as to make generated images more realistic, the discriminator needs to discriminate between real and fake images. The whole process can be seen as a game between the generator and the discriminator, and eventually the two networks reach a dynamic equilibrium. In recent years there has been a boom in research toward GAN, with a range of variants of the GAN emerging. Some works [2, 14] are addressing the problems of the vanilla-GAN such as mode collapse and gradient vanishing, while some works [23, 50, 57] is aimed at improving the performance of GAN to generate better quality images and applying GAN to a wide variety of tasks such as image enhancement, cross-domain translation, text-to-image, and so on. Yuan *et al.* [53] and Khosravy *et al.* [24] utilize GAN for model inversion attacks, they leverage a GAN as an image prior to narrow the search space, and can successfully reconstruct even the high-dimensional data(e.g., face images). Wang *et al.* [49] use a GAN-based structure fusing RGB images and depth maps to generate dense depth maps with fine-grained textures in indoor scene. The ability of GANs to learn the spatial distribution of real samples can add a priori

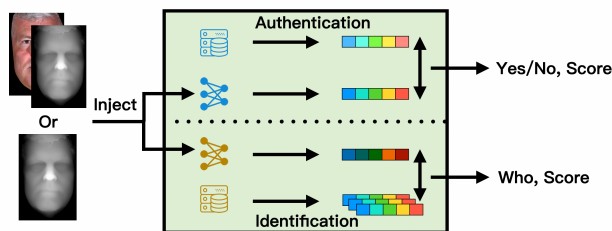

**Figure 1: Attack scenarios: 3D Face Authentication vs. Identification.**

knowledge to a wide variety of tasks, and we suspect that it would be feasible to add a priori knowledge to 3D faces as well.

## 3 THREAT MODEL

In order to evaluate the effectiveness of DREAM and the vulnerability of a given 3D face recognition system in which a DNN-based recognition model, we need to first define the threat model that characterises the adversary.

### 3.1 Attack Scenario

As shown in Fig. 1, we consider two attack scenarios:

- **Face Identification:** multiple 3D template faces belonging to the registered users are recorded in the system. Once a 3D face is fed into the target identification model, the model calculates the similarity between the input face and the template faces, and decides whether the input face is one of multiple template faces based on the similarity score and a predetermined threshold.
- **Face Authentication:** one (or multiple) 3D template face belonging to the registered user is recorded in the system. The model calculates the similarity between the input face and the template face, and decides whether the input face is the same person to the template face.

In this paper, we denote 3D faces in terms of RGB-D image pairs, and these face recognition models use either RGB-D image pair or depth image as input.

### 3.2 Properties for Adversary

**Adversary's goal:** The attacker's goal is to pass the identification or authentication in order to get the control of the device. For example, the attacker obtains the victim's device stealthily and tries to unlock it during his or her short leaves.

**Adversary's knowledge:** We assume the attacker has the following information:

- The attacker holds a RGB photo of the victim, which can be extracted from videos captured by cameras, or downloaded from Internet etc.
- The attacker has a small set of 3D face pairs as an auxiliary dataset, e.g., saved as RGB-D images, which are either public available datasets, or built from 3D scanning. Please note that the auxiliary dataset is non-overlapping with the training dataset and template of the target model.

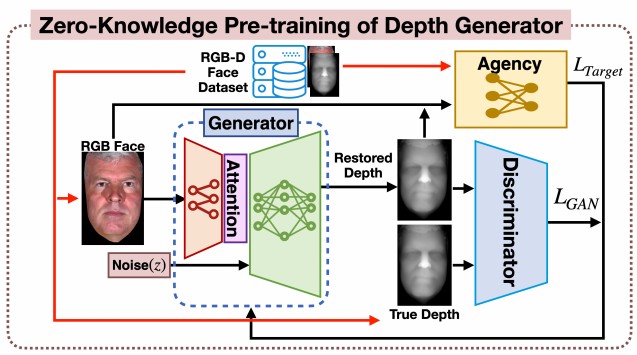

(a) Zero-Knowledge Pretraining of Depth Generator

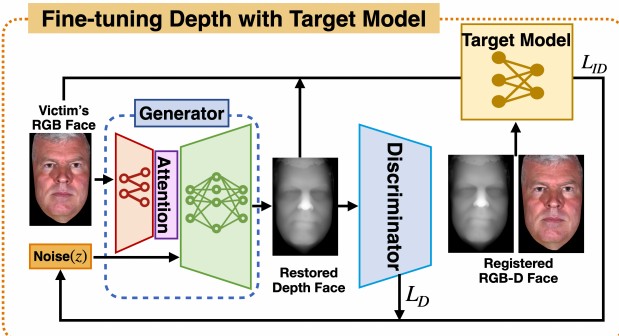

(b) Fine-tuning Depth with Target Model

**Figure 2: Overview of DREAM.**

- The black-box knowledge of the target model, which implies the attacker knows nothing about the target recognition model except its outputs, i.e., similarity scores.

**Adversary's capability:** We assume the attacker has the following capabilities:

- The attacker can inject arbitrary 3D information into a 3D face recognition system, for example, via an optical adversarial attack [30]; for simplicity, we assume that an attacker can inject arbitrary RGB-D images into a black-box 3D face recognition system.
- The attacker can consecutively query the target device for a limited number of times. The target device will be locked (switched off ) temporarily after consecutive failures.
- The attacker can obtain a similar agency model allowing an infinite number of queries. The attacker can obtain the agency model in many ways, such as downloading the same (or similar) recognition applications or using a device of the same type as the target device..

## 4 DESIGN OF DREAM

DREAM is composed of two phases, i.e., *Zero-knowledge Pre-training of Depth Generator* and *Depth Fine-tuning with Target Model*. In the first phase, the attacker first trains a GAN-based generator, which is able to distinguish and recover key depth information to pass 3D face recognition system from 2D face images. Then, the attacker feeds an image of its victim to the generator, and obtains a rough 3D face model. In the second phase, the attackers uses the rough 3D face to query the target model for several times. After each query, the output of the target model is used to optimize the recovered 3D face to a 'better' one. The attack stops when succeeds or reaches to the maximum number of queries allowed. In the following explanation, we consider the problem of reconstructing a depth image from the corresponding RGB image, i.e., restoring depth (D) without loss of generality, since RGB-D images can easily be transformed to other forms of 3D data, such as point clouds.

## 4.1 Zero-knowledge Pre-training of Depth Generator

To reconstruct the depth image, we choose GAN for the reason that it can learn distribution of depth images from public dataset. This adds a priori knowledge in order to generate vaild face depth images. We use a public 3D dataset that has no overlapping identity with the dataset of the target network to train the generator and discriminator. Fig. 2(a) shows the architecture of the offline training.

*4.1.1* **Architecture of Generator.** We use an auxiliary RGB image as an additional input to the generator so that the generated depth image can align with the RGB image and preserve identity-related information, as the RGB image still contains a lot of identity-related information. After extracting the auxiliary information, we add an attention block, Convolutional Block Attention Module(CBAM), which enables the network to automatically learn what and where to pay attention to in an image. CBAM consists of channel attention and spatial attention, applying the channel and spatial attention modules sequentially to learn what and where to pay attention in the channel and spatial dimensions, respectively. The combination of auxiliary information and the attention block enables the GAN to further learn to pay attention to identity-related features based on learning a priori knowledge about the 3D face, and to generate a depth image that can deceive the target recognition model.

*4.1.2* **Dual Contrastive Loss.** The traditional GAN loss is replaced by dual contrastive loss proposed in [52] to train the GAN, which enhance the discriminative ability of the discriminator on small datasets. At the same time, the discriminator pushes the generator to improve the synthesis ability to make the image more real.

The loss function of a conventional GAN is shown in Eq. 1:

$$\min_G \max_D L(G, D) = \mathbb{E}_{x \sim p_{data}(x)}[logD(x)]$$
$$+ \mathbb{E}_{z \sim p_{noise}(z)}[log(1 - D(G(z)))] \quad (1)$$

where $G$ denotes the generator, $D$ denotes the discriminator, $p_{data}$ is the true sample distribution, and $p_{noise}$ is the noise distribution. The process of GAN training can be described as the process of

solving the minimax problem, which can serve two purposes at the same time, the first one is to enable the generator $G$ to generate real samples, and the second one is to enable the discriminator $D$ to better distinguish real samples from the generated samples. However, conventional GAN suffer from problems such as model collapse and gradient vanishing during the training process, and some improved methods [2, 14] solve the above problems starting from the loss function.

Adversarial training relies on the discriminator's ability on real vs. fake classification. As in other classification tasks, discriminators are also prone to over-fitting when the dataset size is limited. The auxiliary dataset used by the attacker to train the GAN is also limited, but the training data within a batch is not fully utilised during training. Inspired by contrastive learning, we treat the ground truth depth image within the same batch as a positive example and the generated depth image as a negative example to drive the discriminator to learn good features or representations by comparing similarities or differences between data samples. We use dual contrastive loss to replace the original GAN loss, the formulations are as eq. 2 and eq. 3:

$$L_{GAN}^+ = \frac{1}{N} \sum_{i=1}^{N} \left[ log \frac{e^{D(depth_i)}}{e^{D(depth_i)} + \sum_{j=1}^{N} e^{D(G(rgb_j, z_j))}} \right] \quad (2)$$

$$L_{GAN}^- = \frac{1}{N} \sum_{j=1}^{N} \left[ log \frac{e^{-D(G(z_j))}}{e^{-D(G(rgb_j, z_j))} + \sum_{i=1}^{N} e^{-D(depth_i)}} \right] \quad (3)$$

where $N$ is the batch-size, $depth_i \sim p_{data}$ is the ground truth depth image, $z_j \sim p_{noise}$ is the noise and $G(rgb_j, z_j)$ is its corresponding generated depth image. For eq. 2, the goal of our loss function is to teach the discriminator how to discriminate a single ground truth depth image out of a batch of generated images. For eq. 3, the discriminator learns to discriminate a single generated image against a batch of ground truth depth images. So the final GAN loss function can be written as Eq. 4.

$$\min_G \max_D L(G, D) = L_{GAN}^+ + L_{GAN}^- \quad (4)$$

*4.1.3* **Target Loss.** We also introduce different target losses according to different target models to make the generated depth image closer to the template face depth image in the target model. Prior to the introduction of target loss, the generator was only able to produce depth image that looked realistic, and in order to be able to fool the target recognition system, it was necessary to make the generated depth image closer to the template depth image after being processed by the target recognition system. So we introduce different target losses according to different target models.

For 3D **face verification**, we introduce the target loss as in eq. 5, which is commonly used in face verification model training.

$$\min_G L_{Target}^{ver}(G) = (1 - Y)\frac{1}{2}(Dis(x, G(rgb, z)))^2$$
$$+ (Y)\frac{1}{2}\{max(0, m - Dis(x, G(rgb, z)))\}^2 \quad (5)$$

where $Y$ denotes whether the two input image pairs match, 0 means match and 1 for not match. $x$ is the ground truth depth image

or RGB-D image pair, $G(rgb, z)$ is is the corresponding generated images. $Dis(x, G(rgb, z))$ is the Euclidean distance between the two input image pairs calculated by the target network, and $m$ is a margin. The purpose is to make the GAN to learn which features are important to pass the system verification.

For 3D **face identification**, we introduce the cross-entropy loss and Eq. 6 as the target loss for different identification model. 3D face identification systems are commonly trained using softmax layer and cross-entropy loss. However, some systems are employed by removing the softmax layer and calculating the cosine similarity between the features to determine which person in the system the input is. So we introduce Eq. 6 as the target loss:

$$\min_G L_{Target}^{ide}(G) = -\frac{1}{N} \sum_{i=1}^{N} log \frac{exp(Sim(x_i, G(rgb_i, z_i)))}{\sum_{j=1}^{N} exp(Sim(x_j, G(rgb_i, z_i)))} \quad (6)$$

where $N$ denotes the batch-size, $x_i$ is the i-th ground truth depth image or RGB-D image pair in a batch, $G(rgb_i, z_i)$ is the corresponding generated images, $Sim(\cdot, \cdot)$ is the similarity between the two input image pairs calculated by the target network. During the time of training, pairs of auxiliary RGB and real depth image are used to train, after inputting the RGB image and noise $z$ to generator, it will generate a corresponding depth image. Then the similarity calculated by target model between real images and generated images is considered as positive examples. The similarity between real images and other generated images in the batch is considered as negative examples. This ensures that similar data samples are close to each other in the target system feature space and dissimilar data samples are far away from each other.

The final loss function can be written as Eq. 7:

$$\min_G \max_D L(G, D) = L_{GAN}(G, D) + \lambda L_{Target} \quad (7)$$

## 4.2 Depth Fine-tuning with Target Model

As shown in Fig. 2(b), after training, our goal is to find a potential vector that enables the generated depth image or image pair (the auxiliary RGB and generated depth image) to achieve the highest similarity with the template face under the target network. We introduce following optimization problem (Eq. 9) to find the optimal vector $\hat{z}$.

$$L_D(z) = -log(D(G(z))) \quad (8)$$

$$\hat{z} = arg \min_z L_D(z) + \alpha L_{ID}(z) \quad (9)$$

where $L_D(z)$ penalizes abnormal face features and target loss $L_{ID}(z)$ encourages the generated depth images to achieve maximum similarity with template face under the target network.

For 3D face verification, the $L_{ID}^{ver}$ is same as $L_{Target}^{ver}$. Because in face verification, the output of target model is the distance between the template and input, the attacker only needs to keep approaching this template.

For 3D face identification, we use cross-entropy loss and cosine embedding loss(Eq. 10) as the $L_{ID}$ for different identification model.

$$L_{ID}^{ide} = \begin{cases} 1 - cos(x_1, x_2), & label = 1 \\ max(0, cos(x_1, x_2) - margin), & label = -1 \end{cases} \quad (10)$$

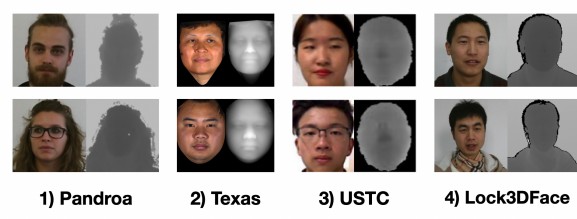

**1) Pandroa    2) Texas    3) USTC    4) Lock3DFace**

**Figure 3: Samples from Datasets.**

If the target system recognizes the generated depth image or image pair (the auxiliary RGB and generated depth image) as the person we want to impersonate, the label is 1, otherwise it is -1, $cos(x_1, x_2)$ is the cosine similarity of the target model output. DREAM will continue to optimise according to the losses until the attack is successful or the number of attempts is exhausted.

## 5 EXPERIMENTS

In this section, we will introduce the experimental settings and demonstrate the effectiveness of DREAM against different FR systems under black-box scenario.

### 5.1 Datasets

We use four datasets, samples are shown in Fig. 3, to evaluate DREAM:

1) *Pandora* [5] uses the Microsoft Kinect One device to acquire depth data and includes over 250k RGB-D images of 20 subjects, 10 males and 10 females.

2) *RGB-D Facial Dataset under Pose Variation* [20] is captured by the PrimeSense camera in the same indoor scene and contains 24839 RGB-D images of 952 identities, including only young Asians, 30% of whom are male and 70% female. In this paper, USTC is referred to this dataset.

3) *Texas 3D* [16] is acquired using a stereo imaging system manufactured by 3Q Technologies (Atlanta, GA) in the x, y, and z dimensions at a very high spatial resolution of 0.32 mm. During each acquisition, color and range images were captured simultaneously, including 1149 pairs of images from 105 adult subjects.

4) *Lock3DFace* [54] is acquired using the second-generation of the Kinect sensor. It totally consists of 5671 RGB-D face video clips belonging to 509 individuals with diverse changes in facial expression, pose, occlusion and time lapse. Among them, 377 are male and 122 are female.

**Pre-processing:** for the first three datasets, we crop these images at the center and resize them to 64×64. 50% and 40% of the images are used to train the face authentication network and our DREAM, respectively, and the rest 10% is used for testing. For lock3DFace, we crop those images at the center and resize them to 224×224, and images from 340 and 100 individuals are used to train the face identification network, and our DREAM, and the rest images of 69 people are used for attacking. It is necessary to ensure that the data for training the target network does not overlap with the data used to train the GAN.

## 5.2 Target Model Implementation

We use three 3D face recognition models to validate our attack.

1) *Face Authentication:* we use Siamese network [43] implemented by PyTorch and a contrasting loss function in Eq. 11.

$$L(W, Y, X_1, X_2) = (1 - Y)\frac{1}{2}(F_W)^2 + (Y)\frac{1}{2}\{max(0, m - F_W)\}^2 \quad (11)$$

where $X_1, X_2$ are the two inputs (RGB-D image pair) to the authentication network $F$, $W$ are the parameters of the authentication network, $F_W$ is the final output, that is, the Euclidean distance between the two inputs under the target network. The optimizer for training is Adam with learning rate 0.001 and batch size 32.

We use false acceptance rate (FAR) and false rejection rate (FRR) to evaluate the authentication model and choose a threshold empirically. FRR is the probability of treating the same person as different persons, but FAR indicates the probability of treating different persons as the same person. FAR and FRR vary with threshold as Fig. 4, we set a threshold for them respectively as Tab. 1, corresponding to the case where FAR+FRR is the smallest.

**Table 1: Threshold Selection of Face Authentication Model.**

|  | **Pandora** | **Texas** | **USTC** |
|---|---|---|---|
| **Threshold** | 1.00 | 0.60 | 0.23 |
| **FAR/FRR** | 0.03/0.03 | 0.07/0.21 | 0.14/0.21 |

2) *Face Identification:* we use Led3D [35] and the model proposed by Uppal *et al.* [47]. Led3D utilizes a Softmax layer with the cross entropy loss to guide network training, but during testing, it discards the Softmax layer and calculates the cosine similarity between the input sample and all the templates in the gallery, taking the identity of the template with the largest similarity. We use pre-trained Led3D to evaluate our attack. We implement Uppal's model using parameter settings mentioned in the literature [47]. During training, one main output and two auxiliary outputs of the model are used with Softmax layer and cross entropy loss, while the two auxiliary outputs are discarded during testing. We evaluate the performance of the two models using rank-one accuracy on Lock3DFace, the results are shown in Tab. 2.

**Table 2: Performance of Two Identification Models.**

| **Model** | **Input** | **Accuracy** |
|---|---|---|
| **Led3D [35]** | Depth | 80.12% |
| **Uppal [47]** | RGB+Depth | 89.61% |

## 5.3 DREAM Implementation

**GAN:** We implement the GAN using PyTorch. We use the same Adam optimizer ($\beta_1$=0.5, and $\beta_2$=0.999) for the generator and discriminator with a learning rate of 0.001. The batch sizes are 32 and 64 for spoofing identification and authentication, respectively. The value of $\lambda$ will be discussed in Sec. 5.4. In the second stage, we set $\alpha$ = 100 and use the SGD optimizer to optimize the potential vector $z$

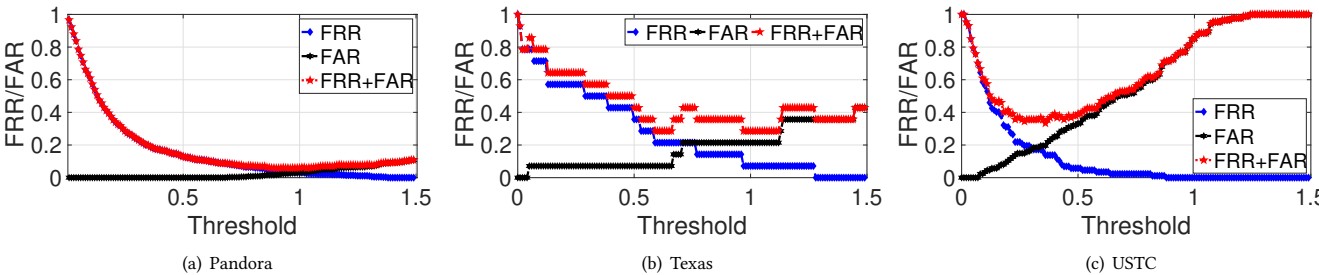

**Figure 4: Relationship between FAR/FRR and Threshold on 3 Datasets.**

with a learning rate of 0.02 and the momentum is 0.9. We randomly initialize $z$ from a zero-mean unit variance Gaussian distribution and optimize it for *five* iterations.

**Attention Block:** We implement CBAM as an attention module using PyTorch. CBAM incorporates two sub-modules, CAM (Channel Attention Module) and SAM (Spatial Attention Module), which add attentional mechanisms on the channel and spatially, respectively.

**Comparisons:** we use three 3D face reconstruction methods, i.e. HRN [27], DECA [11] and Face++ interface [10] , as our comparison baselines. We use pre-trained models published by HRN and DECA to reconstruct faces and generate depth images. For the Face++ interface [10], we need to upload one or more RGB images as required to get a 3D model. After getting the model, we need to render a pair of RGB and depth images. The 3D rendering is implemented using a python script for Blender [45], a free open-source 3D graphics imaging software that runs cross-platform.

### 5.4  Determining Key Parameter $\lambda$.

$\lambda$ is used to balance the target loss and the loss of the GAN itself, making the restored 3D face be able to pass the authentication or identification, and the reconstructed depth image maintain facial geometric structures, simultaneously. We experimentally examine the ASRs under different values of $\lambda$. Tab. 3 shows that the attack against face authentication and identification models are the most effective when $\lambda$ = 1.2 and 1.0, respectively. In our experiments, we set $\lambda$ as 1.2 and 1.0 when the attack targets are face authentication and identification models, respectively.

**Table 3: Impact of $\lambda$ on ASRs**

|  | $\lambda$=0.8 | $\lambda$=1.0 | $\lambda$=1.2 | $\lambda$=1.5 |
|---|---|---|---|---|
| **Pandora** | 73.17% | 81.57% | 94.73% | 94.73% |
| **Texas** | 70.52% | 78.57% | 85.71% | 71.42% |
| **USTC** | 86.20% | 88.50% | 89.65% | 89.65% |
| **Lock3DFace(Led3D)** | 28.78% | 36.36% | 31.81% | 30.31% |
| **Lock3DFace(Uppal)** | 89.32% | 89.32% | 89.32% | 89.32% |

### 5.5  Experimental Results

**Performance Comparison.** The overall performance of DREAM are shown in Tab. 4. It can be seen that DREAM always outperforms 3D face reconstruction attack, and the ASR is much higher than that

of face reconstruction attack on Texas and Lock3DFace (Led3D). This is due to the fact that 3D face reconstruction attack is a one-shot attack, which relies heavily on the reconstruction effect and fails to exploit the weakness of the authentication system. DREAM adds the target loss of the target network during the training process, so that the depth image generated by the generator keeps decreasing the distance from the template, thus the system is misclassified.

The reason why 3D face reconstruction attack has no effect on Led3D is because Led3D only utilizes depth image as input. Furthermore, the 3D reconstruction does not utilize the output of the target model for optimization, resulting in reconstructed depth image that lack sufficient identity-related features. On the contrary, DREAM achieved a maximum ASR of 36%, much greater than the 0 achieved by the 3D face reconstruction attack.

**Table 4: ASRs Comparison of Four Methods**

| Method | Datasets | | | | |
|---|---|---|---|---|---|
| | Pandora | Texas | USTC | Lock3DFace | |
| | | | | Led3D | Uppal |
| Face++ | 55.26% | 3.57% | 38.50% | 0% | 89.32% |
| HRN | 60.52% | 17.85% | 44.82% | 0% | 89.32% |
| DECA | 63.15% | 21.42% | 48.27% | 0% | 89.32% |
| **DREAM** | **94.73%** | **85.71%** | **88.50%** | **36.36%** | **89.32%** |

3D face reconstruction attack achieves comparable performance to DREAM against Uppal [47], this is because the identification model uses the depth image to generate the attention map, and the final features used to make a decision are the RGB features multiplied by the attention map.

We also measure the ASRs for different number of queries. As shown in the Fig. 5, "Auth." denotes the face authentication model, even the first round of queries has a high ASR, as the number of query rounds increases, the ASR also increases, especially for Texas. This shows that both phases of DREAM contribute significantly to the effect of the attack.

The generated depth images and 3D reconstructed depth images for several cases of successful attacks are shown in Fig. 6. The leftmost number is the ID of the case, RGB stands for the auxiliary data that can be utilised, Depth stands for the ground truth depth image, and the others are four of the comparison methods. HRN, DECA and Face++ 3D face reconstruction algorithms are based on face models such as FLAME [29] and BFM [12] for reconstruction.

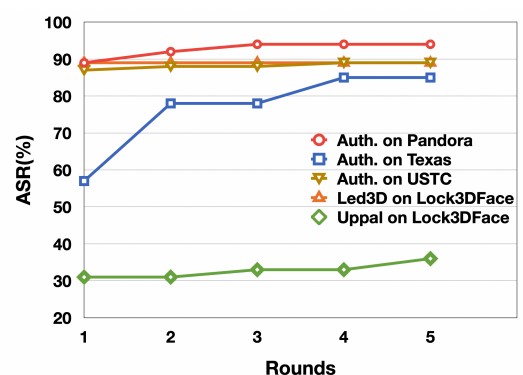

Figure 5: ASRs at Different Rounds

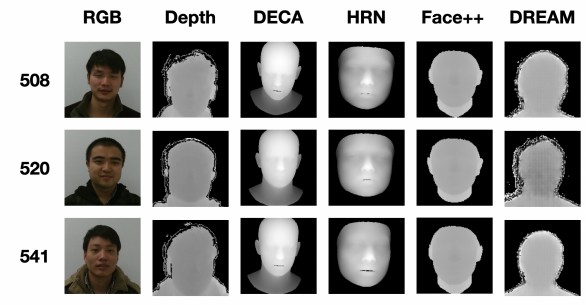

Figure 6: Restored Depth Images of Different Methods.

As a result, they render a depth image that is visually closer to a natural human face. However, 3D face reconstruction is a one-shot deal and cannot exploit the output of the face recognition model to continue adjusting the depth image. DREAM focuses on the behaviour of target recognition models from the ground up, learning which features are better at spoofing target recognition models and continuously optimising them during online attacks.

**Cross-dataset Testing.** We perform cross-dataset attack on the face authentication model. The results are shown in the Tab. 5, for example, 71.26% represents ASR by the attack model trained on Pandora dataset against the target model trained on USTC dataset. The attack model trained on Texas dataset gets a low ASR against the target model trained on Pandora, this is because the distributions of the two datasets may vary significantly, either in terms of the number of individuals or the total number of images, the situation is similar in the opposite case. In summary, the generality of DREAM is good, because we aim to reduce the distance between input and template.

**Ablation Experiment.** To verify the effectiveness of the architecture of GAN, dual contrastive loss and target loss, we performed ablation experiments on four datasets. Tab. 6 shows the change in ASR with the use of each component, which proves that each component contributes to the improvement of the ASR.

Regular GAN just randomly generates a face depth image, if there is a target loss there is an additional goal to make the generated

Table 5: ASRs Cross Datasets

|  | **Pandora** | **Texas** | **USTC** |
|---|---|---|---|
| **Pandora** | 94.73% | 14.28% | 71.26% |
| **Texas** | 44.73% | 85.71% | 83.90% |
| **USTC** | 78.94% | 71.42% | 88.50% |

Table 6: Ablation Experiment

| Method | Datasets | | | | |
|---|---|---|---|---|---|
| | Pandora | Texas | USTC | Lock3DFace | |
| | | | | Led3D | Uppal |
| G | 65.78% | 42.85% | 57.47% | 0% | 89.32% |
| G + $L_T$ | 81.57% | 71.42% | 78.16% | 21.23% | 89.32% |
| G + A + $L_T$ | 86.84% | 78.57% | 83.90% | 27.28% | 89.32% |
| G + A + $L_T$ + $L_{DCL}$ | 94.73% | 85.71% | 88.50% | 36.37% | 89.32% |

G denotes GAN.

$L_T$ denotes target loss.

A denotes attention block.

$L_{DCL}$ denotes dual contrastive loss.

depth image spoof the target recognition. With an auxiliary input and attention block, DREAM can learn to generate key features more accurately from focused areas, rather than blindly generating key features. Dual contrastive loss makes the discriminator more powerful and at the same time forces the generator to improve its generation ability to make the generated image closer to the true depth image distribution, which also helps to the improve the ASR.

In the case where the available auxiliary dataset is small, the modified target loss (Eq. 6) based on the contrastive learning loss can keep the depth image generated based on the auxiliary RGB close enough to the identity of the RGB and far enough from other identities in Led3D. We also use the cross-entropy loss and the cosine embedding loss to train the attack model on Led3D, the cosine similarity between the generated depth image and all other identities is higher than 0.8, including the target identity we want to impersonate. But the cosine similarity between the generated depth and the target identity is not highest, both of which fail to keep the generated depth image sufficiently distant from other identities.

## 6 CONCLUSION

In this paper, we propose a block-box attack method that can fool the 3D face recognition system with depth information recovered from 2D images. The attacker exploits the output of the target model and the 2D images of victim to recover the depth, which is convenient compared to previous methods of recovering depth information. We add target loss of target network and attention block to the GAN to recover the depth of able to pass the authentication, rather than approaching ground truth. Dual contrastive loss also contributes to the ASR. Finally we evaluate the effectiveness of DREAM on four public 3D face datasets. Our experimental results show that DREAM is effective and performs well on different datasets, some 3D face recognition systems still rely on RGB features which is vulnerable.

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
