# OpenReview forum: "Fooling 3D Face Recognition with One Single 2D Image"
_acmmm.org/ACMMM/2024/Conference — MM2024 Oral_

### Official Review · Reviewer_1N8M · 2024-05-20

**Rating:** 4
**Confidence:** 4

**Summary:**

This paper introduces a depth recovery attack method(DREAM), which fools the 3D face recognition system by using depth information recovered from one single 2D image for a limited number of times. Unlike existing 3D face reconstruction attack method that is a one-shot deal, DREAM can continuously optimize model with the target loss. Furthermore, DREAM adds an attention block into generator to extract as much identify-related information as possible with an auxiliary image, and it uses dual contrastive loss to decrease over-fitting with the limited dataset size. According to the results of the experiment, DREAM has a better performance compared to 3D face reconstruction on both 3D face verification and 3D face identification.

**Strengths:**

The author thoroughly analyzes the advantages and disadvantages of existing 3D face reconstruction attacks and then introduces DREAM, which combines the strengths of previous methods while addressing their shortcomings.

Additionally, the author develops effective strategies to overcome challenges in the proposed attack scenario.

**Limitations:**

1. In the introduction, there is a typo in the serial number.
2. In section 4.1.2, the description of the traditional GAN loss is redundant.
3. In Fig. 4, the curve for Texas appears unnatural, and the author does not provide an explanation for this phenomenon.
4. In this experiment, the author trains and tests the model on the same dataset, making it difficult to ensure that the training and test sets are not correlated.
5. In the cross-dataset testing experiment, the results for different datasets are lower than those for the same dataset, and the author does not experiment to explore these results, why performs poor under the cross-dataset setting. I also don't agree that  the generality
of DREAM is good.

**Suitability:**

3

---

### Official Review · Reviewer_fUqu · 2024-05-25

**Rating:** 4
**Confidence:** 3

**Summary:**

The paper introduces an attack method called DREAM, which aims to deceive 3D face recognition models using a reconstructed 3D face from single RGB image from the victim. Unlike other 3D reconstruction techniques that are not specifically designed for identity similarity measurement by a 3D face recognition system, the proposed method is tailored for this purpose through a two-phase optimization scheme. The first phase involves zero-knowledge pre-training of a depth generator using a surrogate 3D face recognition model. The second phase involves fine-tuning with the "black-box" target model, meaning that only similarity scores can be obtained, with limited trial attempts. Experimental results demonstrate the effectiveness of the proposed attack method, achieving an attack success rate of up to 94% for face authentication models and 36% for cross-dataset face identification models.

**Strengths:**

1. The paper clearly introduces a threat model to explain the possible attack scenarios.

2. The design of the two-phase optimization scheme and its corresponding loss functions is novel.

3. The proposed attack achieves a high success rate, even in cross-dataset scenarios, outperforming other conventional 3D reconstruction methods from 2D images.

**Limitations:**

1. Instead of stating that "building 3D face models requires specialized 3D scanning equipment, which incurs high cost" (lines 68-69), the paper should emphasize that it is difficult for an attacker to obtain such models. In contrast, 2D images of the victim are easier to obtain, especially from social networks.

2. The paper mentions that "Statistical model fitting requires massive 3D faces of different persons to obtain a statistical 3D facial model to which small modifications are applied according to a 2D facial image to be reconstructed" (lines 85-88) and "learning the 2D-to-3D mapping through deep neural networks (DNN) is a promising way" (lines 91-95). However, most DNN-based approaches also require a large number of 3D faces for training such models.

3. The assumption that "the attacker can obtain a similar surrogate model allowing an infinite number of queries" means that the attack is not entirely a black-box but rather a gray-box attack. A black-box attack assumes that the model's architecture, weights, and training data are unknown and different from the surrogate model. The experimental design does not clearly indicate the separation of the face recognition model's architecture between training and testing.

**Suitability:**

3

---

### Official Review · Reviewer_UAGJ · 2024-05-26

**Rating:** 2
**Confidence:** 3

**Summary:**

The paper proposes a novel attack method called DREAM to deceive 3D face recognition systems using a single 2D image of the victim's face. The method comprises two main stages: 1) Training a GAN to generate an approximate 3D face model from the 2D image by learning key depth features relevant to the target recognition model, and 2) Iteratively querying the black-box target model and optimizing the generated 3D face to maximize the similarity score and evade recognition. The proposed method is evaluated against both face authentication and identification models on four public datasets.

**Strengths:**

1. The paper addresses the important problem of the vulnerability of 3D face recognition systems to spoofing attacks using 2D images, which are more easily accessible to attackers compared to 3D scans. This highlights the potential security risks associated with such systems.

2. The proposed DREAM framework introduces a novel approach by combining a GAN with attention mechanisms and dual contrastive loss to generate rough 3D faces optimized for the target recognition model. The iterative refinement of the generated faces via black-box queries is an interesting technique that exploits the target model as an oracle.

3. The attack method is evaluated on both authentication and identification tasks using four datasets and three state-of-the-art 3D face recognition models as targets. The results demonstrate relatively high attack success rates, surpassing the performance of reconstructing complete 3D faces in some cases.

4. The paper provides ablation studies to analyze the impact of each component of DREAM, shedding light on their individual contributions to the overall performance.

**Limitations:**

1. The justification for the specific choice of techniques (e.g., dual contrastive loss) over other potential methods is not thoroughly discussed. The paper mentions that the discriminator is prone to overfitting due to the limited size of the dataset, which is why contrastive loss was used. However, it does not explain why contrastive loss was chosen over other regularization techniques that could address the same issue. A comparative analysis with these alternative methods would strengthen the justification for the chosen approach.

2. The paper lacks an explanation of the acronym ASR, which stands for Attack Success Rate. Including this clarification would improve the clarity and comprehension of the results section.

3. Multiple experiments in Table 4 and Table 6 show the same ASR value for the Uppal dataset. Please provide a clear answer as to why the same ASR values occurred.

4. The impact of the proposed attack is not fully explored in real-world scenarios. The experiments focus on digital attacks, but the practicality of fabricating physical masks from the generated depth maps is not investigated. The paper does not discuss the potential challenges and limitations of executing the attack in physical settings, which is crucial for assessing its real-world implications.

**Suitability:**

2

---

### Meta-Review · Area_Chair_TKyr · 2024-07-03

**Recommendation:** Accept (Oral)
**Confidence:** 4

**Metareview:**

The paper introduces an attack method called DREAM, which aims to deceive 3D face recognition models using a reconstructed 3D face from single RGB image. In general, all concerns raised by the reviewers have been addressed and further clarifications provided.
A major concern addressed by the reviewers is the reproducibility of the experimental results, which is often a critical factor in research papers. However, this concern could be mitigated by making the code and data used available to the scientific community.